# Enterovirus 71 VP1 Protein Regulates Viral Replication in SH-SY5Y Cells via the mTOR Autophagy Signaling Pathway

**DOI:** 10.3390/v12010011

**Published:** 2019-12-20

**Authors:** Zi-Wei Liu, Zhi-Chao Zhuang, Rui Chen, Xiao-Rui Wang, Hai-Lu Zhang, Shu-Han Li, Zhi-Yu Wang, Hong-Ling Wen

**Affiliations:** 1Key Laboratory for Infectious Disease Control and Prevention, Department of Microbiological Laboratory Technology, School of Public Health, Shandong University, Jinan 250012, China; 2Department of pathogenic microbiology, Tianjin Center for Disease Control and Prevention, Tianjin 300000, China; dennis.eyre@hotmail.com

**Keywords:** EV71, VP1, autophagy, virus replication

## Abstract

Background: Enterovirus 71 (EV71) is the main pathogen that causes severe hand, foot, and mouth disease with fatal neurological complications. However, its neurovirulence mechanism is still unclear. Candidate virulence sites were screened out at structural protein VP1, but the function of these candidate virulence sites remains unclear. Several studies have shown that autophagy is associated with viral replication. However, the relationship between VP1 and autophagy in human neurons has not been studied. Methods: A recombinant virus—SDLY107-VP1, obtained by replacing the VP1 full-length gene of the SDLY107 strain with the VP1 full-length gene of the attenuated strain SDJN2015-01—was constructed and tested for replication and virulence. We then tested the effect of the recombinant virus on autophagy in nerve cells. The effect of autophagy on virus replication was detected by western blot and plaque test. Finally, the changes of mTOR signaling molecules during EV71 infection and the effect of mTOR on virus replication at the RNA level were detected. Results: Viral recombination triggered virulence attenuation. The replication ability of recombinant virus SDLY107-VP1 was significantly weaker than that of the parent strain SDLY107. The SDLY107 strain could inhibit autophagic flux and led to accumulation of autophagosomes, while the SDLY107-VP1 strain could not cause autophagosome accumulation. The synthesis of EV71 RNA was inhibited by inhibiting mTOR. Conclusions: Replacement of VP1 weakened the replication ability of virulent strains and reduced the level of autophagy in nerve cells. This autophagy facilitates the replication of virulent strains in nerve cells. VP1 is an important neurovirulence determinant of EV71, which affects virus replication by regulating cell autophagy. mTOR is a key molecule in this type of autophagy.

## 1. Introduction

Enterovirus 71 (EV71), which belongs to the Picornaviridae family, is an icosahedral non-enveloped single-stranded RNA virus and is the second most important neurotropic enterovirus [1]. It causes hand, foot, and mouth disease (HFMD), which occurs mainly in young children below 4 years of age and usually presents as a self-limiting, mild febrile disease with symptoms including ulcers in the mouth, maculopapular rashes, or blister-like eruptions on the palms and soles [2]. EV71 has been increasingly associated with neurological disorders that range from aseptic meningitis with or without pulmonary edema to brain stem encephalitis and poliomyelitis-like acute flaccid paralysis, particularly among infants. EV71-related hand, foot, and mouth disease outbreaks have been reported throughout the world, including in Singapore, Malaysia, Vietnam, Cambodia, China, Australia, and Japan [3,4,5,6,7,8]. Unfortunately, there are no effective prophylactic or therapeutic agents against EV71, although recent progress has been made.

A large precursor protein is encoded by the RNA genome of EV71, which is then processed by viral proteases into structural (VP1, VP2, VP3, and VP4) and nonstructural (2A, 2B, 2C, 3A, 3B, 3C, and 3D) proteins [9]. The structural proteins constitute the viral capsid, and the nonstructural proteins participate in virus replication. Among the four structural proteins, VP1 is the most external, surface-accessible, and immunodominant protein among the picornavirus capsid proteins [10]. Virus–receptor binding ability and virulence are affected by VP1 mutations, which even allow the virus to escape the host immune response [11,12,13,14]. EV71 VP1 protein activates calmodulin-dependent protein kinase II, which phosphorylates the N-terminal domain of vimentin on serine 82. Vimentin phosphorylation and rearrangement may enhance EV71 replication by playing structural roles in the formation of the replication factories [15]. In addition, human annexin II protein can bind to EV71 VP1 via VP1 amino acids 40–100, a region different from the known receptor-binding domain, thereby enhancing EV71 replication [16]. Effective cleavage of VP0 precursors is regulated by the amino acid at position 107 of VP1 during EV71 assembly. Virus growth kinetics are significantly reduced due to mutation of amino acid at position 107 of VP1 [17]. EV71 replication is inhibited by miR-2911, a honeysuckle-encoded atypical microRNA targeting the VP1 gene [18]. In conclusion, the VP1 protein plays an important role in the replication cycle of EV71.

Autophagy is a degradative process that degrades protein aggregates and damages organelles. It is monitored by the conversion of the microtubule-associated protein 1A/B-light chain 3 (LC3) from its cytosolic resting state (LC3-I) to the lapidated form LC3-II. Protein p62 is considered a marker for autophagy-mediated protein degradation or autophagic flux. Many pathogens interact positively or negatively with the host autophagic pathway. The release of EV71 virions can be inhibited by inhibiting autophagy in RD cells [19]. Autophagy was induced in EV71-infected NSC-34 cells, which facilitated viral replication and non-lytic exit [20]. Mouse Schwann cell autophagy was promoted by EV71 VP1, which can up-regulate endoplasmic reticulum stress [21]. However, the relationship between VP1 and autophagy in human nerve cells has not been studied.

The mammalian target of rapamycin (mTOR) is a key homeostatic regulator of cell growth, proliferation, survival, and metabolism via upregulating protein and lipid synthesis and inhibiting excessive autophagy. In mammalian cells, the phosphatidylinositol 3-kinase (PI3K)/Akt/mTOR signaling pathway is the primary pathway that regulates autophagy when cells are exposed to certain conditions, such as starvation, oxidative stress, infection, and tumor suppression. mTOR is closely related to autophagy induced by viral infection. Replication of the porcine epidemic diarrhea virus (PEDV) is promoted by rapamycin, an autophagy inducer that inhibits the mTOR signaling pathway [22]. Inhibition of mTOR by rapamycin promotes autophagy and viral mRNA replication, but does not affect VP1 expression during CVB3 infection [23]. Inhibition of mTOR promotes viral replication. However, the relationship between mTOR and EV71 replication is unclear.

In this study, the recombinant virus SDLY107-VP1 obtained by replacing the full-length VP1 gene of the SDLY107 strain with the full-length VP1 gene of the attenuated strain SDJN2015-01 was constructed to explore the effect of VP1 on viral replication. On this basis, the relationship between VP1 and autophagy in viral replication was explored, and the relationship between EV71 replication and mTOR was further explored. This research provides new ideas for the study of the pathogenesis of EV71 and the development of antiviral drugs.

## 2. Materials and Methods

### 2.1. Antibodies and Chemical Reagents

The antibodies used in the study were anti-LC3B (Boston, USA, Cell Signaling Technology, Cat#2775), anti-p62 (Cambridge, UK, Abcam, 56416), anti-enterovirus 71 (Abcam 169442), anti-βactin (Beijing, China, ZSGB-BIO), anti-GAPDH (Wuhan, China, Proteintech, 60004), anti-mTOR (Cell Signaling Technology Cat#2983), anti-phospho-mTOR (Boston, USA, Cell Signaling Technology, Cat#5536), FICT-conjugated Affinipure Goat Anti-Mouse IgG (ZSGB-BIO), HPR-conjugated Affinipure Goat Anti-Rabbit IgG (Proteintech, SA-00001-2), and HPR-conjugated Affinipure Goat Anti-Mouse IgG (Proteintech, SA-00001-1). The anti-EV71 antibody used in the immunofluorescent assay was a mouse-derived polyclonal antibody made by our group [24]. Chemical reagents used in the study were rapamycin (Sigma, Germany, V900930), 3-Methyladenine (3-MA, Sigma, M9281) and chloroquine (CQ, Sigma, C6628).

### 2.2. Cell Lines and Viruses

Rhabdomyosarcoma (RD) cells were provided by the Shandong Center for Disease Control (Purchased from ATCC, USA) and cultured in minimum essential medium (MEM, USA, Gibco) supplemented with 10% heat-inactivated fetal bovine serum (FBS, Gibco) at 37 °C in the presence of 5% CO_2_. Human neuroblastoma (SH-SY5Y) cells provided by the Department of Biochemistry and Molecular Biology, School of Basic Medicine, Shandong University (Purchased from ATCC, USA) were cultured in MEM medium (Gibco) supplemented with 10% heat-inactivated fetal bovine serum (FBS, Gibco) at 37 °C in the presence of 5 % CO_2_. The severe EV71 strain SDLY107 was isolated from a fatal case of HFMD from Linyi City, Shandong Province, China in 2011 [25]. In vivo and in vitro experiments showed that this strain has faster replication and stronger pathogenicity compared to strains isolated from patients with a mild illness [26,27]. The full-length infectious cDNA clones of the SDLY107 strain (SDLY107^RV^) were successfully constructed using reverse genetics technology, and no significant differences in biological characteristics were observed between wild-type SDLY107 and rescued SDLY107^RV^ [28]. The mild EV71 strain SDJN2015-01 was isolated from a child with mild hand, foot, and mouth disease and no neurological complications. The recombinant EV71 virus SDLY107-VP1 was rescued by chimeric infectious clone pMD19T-107-VP1, which was constructed by replacing the VP1 full-length fragment of SDLY107 with the VP1 gene of SDJN2015-01. The target fragment to be replaced was amplified by PCR and ligated to the vector to obtain a recombinant plasmid containing the DNA fragment of the VP1 sequence of the SDJN2015-01 strain. The recombinant plasmid and the full-length plasmid of pMD-19T-107 preserved in the laboratory were identified by restriction endonuclease Nru I and BsiW I for restriction endonuclease digestion. The DNA fragment containing the VP1 sequence of the SDJN2015-01 strain and the DNA fragment of the remaining sequence of the SDLY107 strain except the VP1 sequence were then recovered. The recovered products were transformed after T4 connection to obtain a full-length DNA fragment of SDLY107-VP1, which was identified and subjected to in vitro transcription to obtain the RNA of SDLY107-VP1. All strains were propagated in RD cells in MEM supplemented with 1% FBS. The virus titer was routinely determined by plaque assay. The virus suspension of different dilutions was inoculated on RD monolayer cells to make the virus adsorb onto the cells, and then covered with a layer of nutrient methylcellulose culture medium. When there was no new plaque, crystal violet fuel was added. Because dead cells cannot uptake crystal violet fuel, the associated plaque will be unstained. In theory, a plaque is formed by infection of a virus particle. The plaque morphology was observed and the number of plaques was counted and multiplied with the dilution multiple to get the virus titer.

### 2.3. Virus Infection

SH-SY5Y cells were infected with SDLY107, SDJIN2015-01, and SDLY107-VP1 at a multiplicity of infection (MOI) of 1. They were then replenished with fresh MEM containing 1% FBS, which was used for the sham group after washing with PBS. For experiments that added inhibitors, cells were pre-treated with the inhibitors (dissolved in MEM or DMSO) for 2 h before viral infection. They were then washed with PBS and infected with the virus for 1 h, and fresh MEM containing 1% FBS for the sham group.

### 2.4. Replication Kinetics

SH-SY5Y cells and RD cells were seeded at a density of 2 × 10^5^ cells per mL in 12 well plates and, when SH-SY5Y reached 70%–80% percent confluence, cells were infected with EV71 strains at MOI = 1. The cells and culture supernatant were sampled together every 12 h post infection. The corresponding viral load at 0 h was the amount of RNA in the prepared virus suspension. Total viral RNA was extracted using an RNA extraction kit (E.Z.N.A.^®^Viral RNA Kit, OMEGA, Guangzhou, China) according to the manufacturer’s instructions. The replication kinetics of SDLY107, SDLY107-VP1, and SDJN2015-01 were determined by real-time quantitative PCR (qRT-PCR) as described in a previous study [29]. Experiments were performed in triplicate, with three repeats for each experiment.

### 2.5. Lactate Dehydrogenase (LDH) Cytotoxicity Assay

SH-SY5Y cells were seeded at a density of 2 × 10^5^ cells per mL in 12 well plates and, when SH-SY5Y reached 70%–80% percent confluence, cells were infected with EV71 strains at MOI = 1. LDH activity was detected using a Cytotoxicity Assay Kit (Shanghai, China, Beyotime,) according to the manufacturer’s protocol. The rate of cell injury was calculated by LDH content as follows: Rate of cell injury (%) =$\frac{\mathrm{absorbance}\;\mathrm{of}\;\mathrm{treatment}\;\mathrm{group}\;-\;\mathrm{absorbance}\;\mathrm{of}\;\mathrm{control}\;\mathrm{group}}{\mathrm{absorbance}\;\mathrm{of}\;\mathrm{maximum}\;\mathrm{enzyme}\;\mathrm{activity}\;\mathrm{group}\;-\;\mathrm{absorbance}\;\mathrm{of}\;\mathrm{control}\;\mathrm{group}}$. Experiments were performed in triplicate, with three repeats for each experiment.

### 2.6. Drugs Used to Induce or Inhibit Autophagy

A measure of 3-MA (50 μM, dissolved in MEM) was used to inhibit the early stage of autophagy. CQ (20 μM, dissolved in MEM) was used to inhibit autophagosome degradation, and rapamycin (10 nM, dissolved in DMSO) was used to promote degradation of autolysosome. Two h later, cells were washed three times and infected with the three strains of the virus. MEM containing 1% FBS was used for the sham group.

### 2.7. Detection of Autophagosomes, Viruses, and mTOR in Cells by Immunofluorescent Assay

SH-SY5Y cells were seeded on glass coverslips at 70%–80%confluency and infected with EV71. The cells were fixed in 4% paraformaldehyde for 10 min at room temperature, and blocked with 5% bovine serum albumin in 0.1% Triton X-100 (diluted in PBS) for 2 h, followed by incubation with the primary antibody overnight at 4 °C. After washing three times, the samples were incubated with secondary antibodies for 60 min, and observed under a fluorescent microscope after three washes. Autophagosomes were detected using the CYTO-ID^®^ Autophagy Detection kit (New York, USA, Enzo) in strict accordance with the instructions. The images were captured using a Carl Zeiss Microscope and processed using the software provided by the manufacturer.

### 2.8. Detection of Autophagy-Related Protein and Virus VP1 Protein by Western Blot

SH-SY5Y cells exposed to various conditions were washed twice with ice-cold PBS, then lysed with RIPA lysis buffer (Beyotime) containing 0.1% phenylmethylsulfonyl fluoride (PMSF, Beyotime), and centrifuged at 15,000 rpm for 10 min at 4 °C. The samples were subsequently boiled and denatured. Next, 30 µg of cell protein samples were subjected to 10%–12% SDS polyacrylamide gel electrophoresis after measuring the protein concentration using the BCA Protein Assay kit (Beyotime Institute of Biotechnology, Shanghai, China), and transferred to 0.22 µm PVDF membranes (Millipore) and then blocked with 5% non-fat milk for 1 h (room temperature). The membranes were incubated with the primary antibody at 4 °C overnight, and incubated with horseradish-peroxidase-conjugated secondary antibodies for 60 min followed by washing. Finally, protein bands were detected using the ultra-sensitive multi-function imager.

## 3. Results

### 3.1. Construction and Rescue of Recombinant Virus SDLY107-VP1

We compared the VP1 amino acid sequences of the SDLY107 and SDJN2015-01 strains, and the results are shown in Figure 1A. There were differences between the 146th (V→I) and 147th (V→A) amino acids of SDLY107 and SDJN2015-01. To explore the role of VP1 mutations in EV71 replication, a recombinant virus replacing the VP1 full-length fragment was constructed and rescued. The construction schematic diagram of SDLY107-VP1 is shown in Figure 1B–D. The VP1 upstream and downstream fragments of the SDLY107 strain and the VP1 fragment of the SDJN2015-01 strain were obtained by PCR reaction (Figure 2A). The recombinant DNA fragments were obtained by fusion PCR using the three DNA fragments as templates (Figure 2B), and linked to the pMD-19T vector for blue and white spot screening. The pMD-19T plasmid containing the recombinant DNA fragment and the full-length pMD-19T-107 plasmid were identified by double digestion with restriction endonucleases Nru I and BsiW I (Figure 2C). The DNA fragments containing the VP1 sequence of the SDJN2015-01 strain were recovered and the remaining fragments of the VP1 sequence of the SDLY107 strain were removed. The recovered products were transformed after T4 connection, and the recombinant plasmid pMD-19T-107-VP1 was successfully obtained and digested by restriction endonuclease XbaI and Hind III. The full-length DNA fragment of SDLY107-VP1 was recovered (Figure 2D) and transcribed in vitro to obtain the RNA of the recombinant virus SDLY107-VP1.The SDLY107 strain and the SDLY107-VP1 strain were all identical except for the VP1 gene.

### 3.2. Recombinant Virus Biological Characteristics

SH-SY5Y cells and RD cells were infected with the SDLY107 strain, SDLY107-VP1 strain, and SDJN2015-01 strain. Virus replication curves were analyzed by detecting the total viral RNA. There were no significant differences among the replication curves of the three strains of virus in RD cells (Figure 3A). However, the replication power of the recombinant virus SDLY107-VP1 was significantly weaker than that of the parent strain SDLY107, especially between 24 h and 84 h (Figure 3B). Replacement of VP1 gene was concluded to reduce viral replication in neurons. Changes in amino acids 146 and 147 were related to the neurovirulence of the virus. SH-SY5Y cells were then infected with the three strains. The rate of cell injury was determined by LDH at 24 h after infection. The rate of cell injury of the recombinant virus SDLY107-VP1 was significantly weaker than that of the parent strain SDLY107 (Figure 3C); replacement of VP1 fragments attenuated the ability of virulent strains to cause cell injury.

### 3.3. Autophagic Flux in SH-SY5Y Cells Is Inhibited by EV71 VP1 Protein

To explore the mechanism by which the VP1 gene affects the replication ability of virus in nerve cells, autophagy, which has been proven to be related to viral replication, was quantitatively analyzed. SH-SY5Y cells were infected with the three strains. Cell lysates were collected at 6 h, 9 h, 12 h, and 24 h. Uninfected cells served as the sham control. LC3 and p62 were examined by western blot analysis. SDLY107 strain infection resulted in an increase in the LC3 and p62 content, but the SDJN2015-01 strain and SDLY107-VP1 strain did not cause significant changes in LC3 and p62 levels (Figure 4A,C,D). This indicated that SDLY107 could inhibit autophagic flux in SH-SY5Y cells. The SDLY107-VP1 and SDJN2015-01 strains did not have a significant impact on autophagic flux in SH-SY5Y cells. Figure 4B and 4E shows the fluorescent staining of autophagosomes in SH-SY5Y cells. The number of autophagosomes in SH-SY5Y cells infected with SDLY107 strain increased with the prolongation of infection time. However, the number of autophages did not increase significantly after infection with SDLY107-VP1 and SDJN2015-01 strains. This indicated that SDLY107 could lead to the accumulation of autophagosomes, which is beneficial for viral replication. The replacement of VP1 attenuated the inhibition of autophagic flow by virulent strains in SH-SY5Y cells.

### 3.4. VP1 Protein Affects Virus Replication Associated with Cell Autophagy

We treated cells with 3-MA 50 μM, CQ 20 μM, and rapamycin10 nM, and the levels of p62 and LC3 in cells were measured after 24 h. The 3-MA caused an increase in intracellular p62 levels and a decrease in LC3 levels (Figure 5A,B). CQ led to an increase in intracellular p62 and LC3 levels (Figure 5C,D). However, rapamycin caused a decrease in intracellular p62 and LC3 levels (Figure 5E,F). Our results indicated that 3-MA can inhibit the formation of autophagosomes, CQ can inhibit autophagic flux and lead to the accumulation of autophagosomes, while rapamycin can promote anautophagic flux and accelerate the degradation of autophagosomes. In order to explore the relationship between autophagy and viral replication, SH-SY5Y cells were treated with 3-MA, CQ, and rapamycin for 2 h and then infected with SDLY107, SDLY107-VP1, and SDJN2015-01, respectively. Cell lysates were harvested after 24h. LC3, p62, and capsid protein VP1 levels were detected by western blot. The culture supernatant was collected after 24 h and subjected to plaque assay to determine the viral titer. After infection with SDLY107 and SDLY107-VP1, the virus content in cell and culture supernatants decreased, and the overall replication level of virus decreased in the rapamycin-treated group. In the 3-MA-treated group, the virus content in cells did not change significantly, but the virus content in culture supernatant decreased, and the overall replication level of the virus decreased. However, in the CQ-treated group, the virus content in cells and culture supernatant increased and the overall replication level of virus increased (Figure 6A,B). Unfortunately, due to the weak replication ability of SDJN2015-01 in nerve cells, no obvious VP1 protein band was detected by western blotting except in the CQ treatment group, but the change of virus titer in the culture supernatant was the same as that of the other two viruses. The results showed that the accumulation of autophagosomes is beneficial to virus replication in SH-SY5Y cells. The replication ability of the SDLY107 strain was stronger than that of the replacement strain SDLY107-VP1 because the SDLY107 strain can lead to the accumulation of autophagosomes. The replacement of VP1 results in the loss of ability of virulent strain to inhibit autolysosome degradation.

### 3.5. Autophagy Affects the Survival of Nerve Cells Infected with EV71

SH-SY5Y cells were treated with 3-MA, CQ, and rapamycin for 2 h and then infected with SDLY107, SDLY107-VP1, and SDJN2015-01, respectively. The rate of cell injury was detected by LDH at 24 h after infection. After infection with three strains, the rate of cell injury was higher in the CQ-treated group than in the control group, whereas the effect of the rapamycin treatment was the opposite. The cell injury rate of the 3-MA treatment group was not significantly different from that of the control group (Figure 6C). This suggests that the accumulation of autophagosomes increased the cell injury rate. The difference of cell injury ability between the SDLY107 and SDLY107-VP1 strains in SH-SY5Y cells was related not only to viral replication but also to autophagy; the SDLY107 strain and the SDLY107-VP1 strain were identical except for the VP1 gene. Therefore, VP1 regulates viral replication and cell injury by affecting the accumulation of autophages.

### 3.6. mTOR Is a Key Molecule Affecting EV71 Replication in Nerve Cells

mTOR signaling is generally involved in regulating cell survival, cell growth, cell metabolism, protein synthesis, and autophagy, as well as homeostasis [30]. SH-SY5Y cells were infected with the three strains, cell lysates were harvested, and mTOR/p-mTOR was detected by western blot. The SDLY107 strain significantly reduced p-mTOR content; however, no similar results were observed for the other two strains (Figure 7A). VP1 replacement attenuated the ability of virulent strains to inhibit mTOR phosphorylation, which is beneficial to autophagy. This autophagy facilitates the replication of virulent strains in nerve cells. VP1 affects virus replication via autophagy related to mTOR signaling molecules. We selected the SDLY107 strain to infect SH-SY5Y cells, and found by immunofluorescence test that EV71 virus and mTOR can co-locate (Figure 7B). This also indicates that there could be interaction between the EV71 virus and mTOR molecule. Subsequently, SH-SY5Y cells pre-treated with rapamycin (10 nM) were infected with SDLY107, and the viral capsid protein VP1 was examined by western blot and intracellular viruses were detected by immunofluorescent assay. The amount of total RNA and RNA in the cell culture supernatant was detected by real-time fluorescent quantitative PCR. Inhibition of mTOR by rapamycin, which can promote the degradation of autolysosomes, can restrain viral RNA synthesis and block viral protein VP1 expression caused by the SDLY107 infection (Figure 7C–E).

## 4. Discussion

A number of virulence determinants for EV71 have been reported, which may influence virus assembly, attachment, replication, or cell tropism. However, the mechanism by which VP1 affects viral replication and virulence has not been elucidated. In this study, a recombinant virus with the VP1 fragment replaced was successfully constructed. The SDLY107 strain and the SDLY107-VP1 strain were identical except for the VP1 gene. Replacement of the VP1 fragment altered viral replication. Increased replication capacities of EV71 strains have previously been associated with increased disease severity in animal models, and positive correlations have been found between virus titers in the respiratory tract and disease severity in humans [29,31,32]. Therefore, the VP1 protein is closely related to the virulence and pathogenicity of the virus. A variety of viruses can manipulate autophagy to assist their replication [33]. Autophagy is induced in host cells by viruses, including human cytomegalovirus, hepatitis C virus, herpes simplex virus-1, CVB3, influenza A virus, HIV-I, and EV71 [34,35,36,37,38,39,40]. EV71 replication induced the formation of autophagosomes and increased disease severity and viral titer [41]. Virus-containing autophagic vacuoles were isolated from the culture supernatant which suggests that autophagy contributed to the release and spread of the virus [20]. Our research demonstrated that SDLY107could inhibit autophagic flux in SH-SY5Y cells and lead to the accumulation of autophagosomes. However, replacement of VP1 directly led to decreased autophagosome accumulation and reduced viral replication. The accumulation of autophagosomes is beneficial to virus replication.

In this study, the viral titer and the rate of cell injury were decreased when autophagic flux was promoted by rapamycin (Figure 6). Host defense is another important role of autophagy in addition to maintaining cell homeostasis. Autophagy improves host defense mechanisms through several biological functions [42,43,44]. The role of autophagy for host defense against viruses strongly depends on the type of infection and involves several mechanisms, especially those involved in autophagosome generation and maturation. RNA viruses seem to stabilize autophagosomes by preventing their degradation. The human immunodeficiency virus type-1 (HIV) blocks autophagosome maturation in infected macrophages [45]. The degradation of autophagosomes and the elimination of the invading pathogens are accelerated when autophagy induced by a virus is promoted. Therefore, the virus titer and cell injury rate were reduced after treatment with rapamycin, but the opposite result was obtained when autophagosome degradation was inhibited (Figure 6).

Although EV71 infection can induce autophagy in vivo or in vitro, its exact mechanism is unclear. The PI3K/Akt/mTOR pathway plays a vital role in the regulation of autophagy, and suppressing mTOR activity leads to the initiation of autophagy and the formation of autophagosomes [46]. This study demonstrated that VP1 replacement attenuates the ability of virulent strains to inhibit mTOR phosphorylation, which is beneficial to autophagy. The formation and accumulation of autophagosomes are beneficial to the replication of the virulent strains in nerve cells. However, inhibition of mTOR activation not only leads to the initiation of autophagy and the formation of autophagosomes, but also promotes the degradation of autolysosomes. The cell injury rate, intracellular viral VP1 protein content, and culture supernatant titer were reduced after cells were treated with rapamycin, which could promote the degradation of autolysosomes. The synthesis of viral RNA was inhibited by rapamycin.

In summary, the study demonstrated that replacement of EV71 VP1 resulted in decreased replication of EV71 in SH-SY5Y cells by affecting cell autophagy via the mTOR pathway. The synthesis of viral RNA was inhibited by rapamycin, which inhibits mTOR and promotes the degradation of autolysosomes. The study provides new insights into the molecular mechanism of EV71 and EV71-related HFMD.

## Figures and Tables

**Figure 1 viruses-12-00011-f001:**
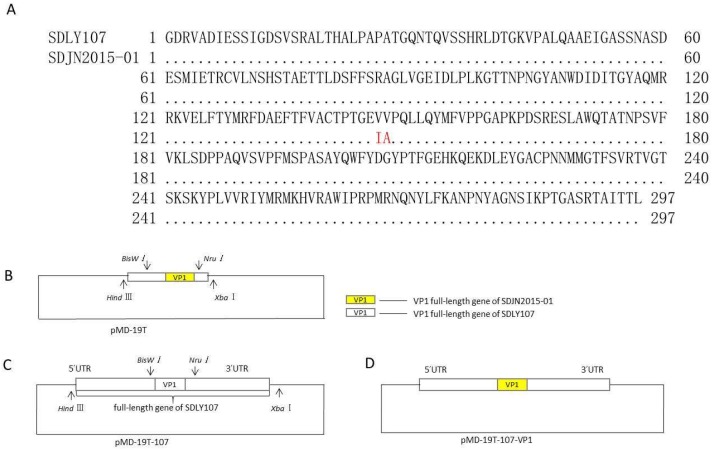
Schematic diagram of SDLY107-VP1. (**A**) The VP1 amino acid sequence comparison results of SDLY107 and SDJN2015-01 strains are shown. (**B**) The upstream and downstream fragments of VP1 of SDLY107 strain and VP1 of SDJN2015-01 strain were used as templates for fusion PCR. The obtained recombinant DNA fragments were linked to the pMD-19T vector and identified by double digestion. (**C**) The full-length pMD-19T-107 plasmid was identified by double digestion, it contains a full-length gene including the untranslated region (UTR) of SDLY107 strain. (**D**) DNA fragments containing the VP1 sequence of the SDJN2015-01 strain and the remaining DNA fragments without the VP1 fragment of the SDLY107 strain were recovered, then transformed after T4 connection, and the recombinant plasmid pMD-19T-107-VP1 was successfully obtained.

**Figure 2 viruses-12-00011-f002:**
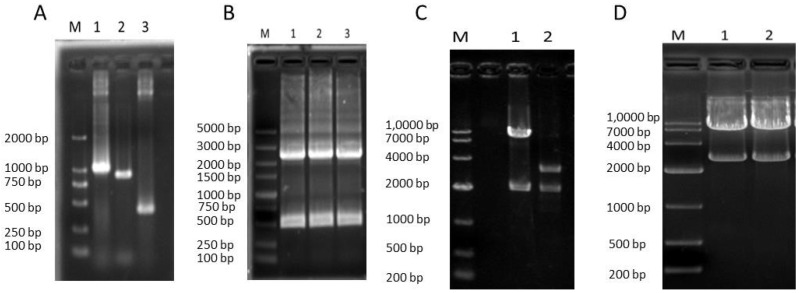
Recombinant virus SDLY107-VP1. (**A**) The PCR product of the 1430–2459 bp upstream fragment of VP1 of the SDLY107 strain, the 3308–3742 bp downstream fragment, and the VP1 fragment of the SDJN2015-01 strain. M is DL2000 DNA Marker; Lane 1 is upstream 1430–2459 bp of VP1 of the SDLY107 strain, and the length is about 1029 bp; Lane 2 is VP1 fragment of the SDJN2015-01 strain, and the length is about 891 bp; Lane 3 is downstream 3308–3742 bp of VP1 of the SDLY107 strain, and the length is approximately 434 bp. (**B**) The fusion products of the 1430–2459 bp upstream fragment, 3308–3742 bp downstream fragment and VP1 fragment of the SDJN2015-01 strain were obtained. M is a DL5000 DNA Marker; Lanes 1–3 are three DNA fragment fusion PCR products, and the length is about 2313 bp. (**C**) Plasmid pMD-19T-107 and pMD-19T-VP1 double digestion enzyme electrophoresis. M is DL1,0000 DNA Marker; Lane 1 is the product of the pMD-19T-107 plasmid after Nru I and BsiW I digestion, the fragment length is about 8.1 kb and 2.0 kb; Lane 2 is the pMD-19T-VP1 plasmid after Nru I and BsiW I digestion. The product fragment length is approximately 3.0 kb and 2.0 kb. (**D**) Plasmid pMD-19T-107-VP1 double digestion enzyme electrophoresis. M is DL1,0000 DNA Marker; Lanes 1 and 2 are the product of the pMD-19T-107-VP1 plasmid digested with XbaI and HindIII, and the fragment length is about 7.4 kb and 2.7 kb.

**Figure 3 viruses-12-00011-f003:**
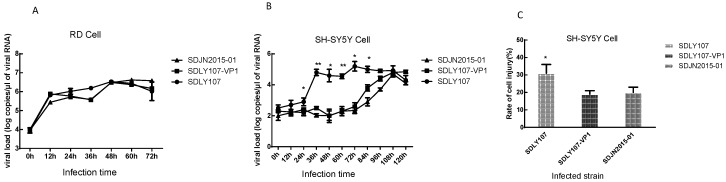
(**A**)The replication curves of the three strains on rhabdomyosarcoma (RD) cells. (**B**) The replication curves of the three strains on SH-SY5Y cells. (**C**)The rate of cell injury of SH-SY5Y cells infected with the three strains at 24 h. The data represent means + SD from three experiments. * *p* < 0.05, ** *p* < 0.001 compared with the SDLY107-VP1 treated group.

**Figure 4 viruses-12-00011-f004:**
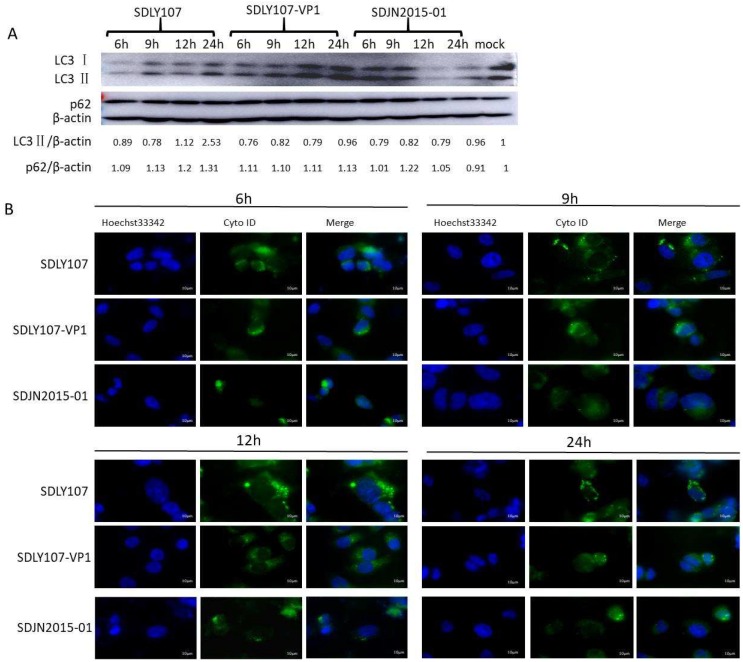
**(A)** Western blot assay for the effect of recombinant virus SDLY107-VP1 strain, parental virus SDLY107 strain, and SDJN2015-01 strain on autophagy. SH-SY5Y cells were infected with EV71. Cell lysates were collected at 6 h, 9 h, 12 h, and 24 h. Uninfected cells served as the sham control. LC3 and p62 were examined by western blot. (**B**) The effects of the recombinant virus SDLY107-VP1 strain, parental virus SDLY107 strain, and SDJN2015-01 strain on autophagy were detected by immunofluorescent staining. Hoechst 33342 blue fluorescent dye stains the cell nucleus blue, and Cyto ID green fluorescent dye stains autophagous bodies green during autophagy. (**C**) The content of p62 protein in SH-SY5Y cells infected with the three viruses. (**D**) The content of LC3 protein in SH-SY5Y cells infected with the three viruses. (**E**) The number of autophages in SH-SY5Y cells infected with the three viruses. Autophagous bodies in each cell were counted, 100 cells per sample. The data represent means + SD from three experiments. * *p* < 0.05, compared with the mock group; # *p* < 0.001, compared with the mock group; ** *p* < 0.05, compared with the SDLY107-VP1-treated group.

**Figure 5 viruses-12-00011-f005:**
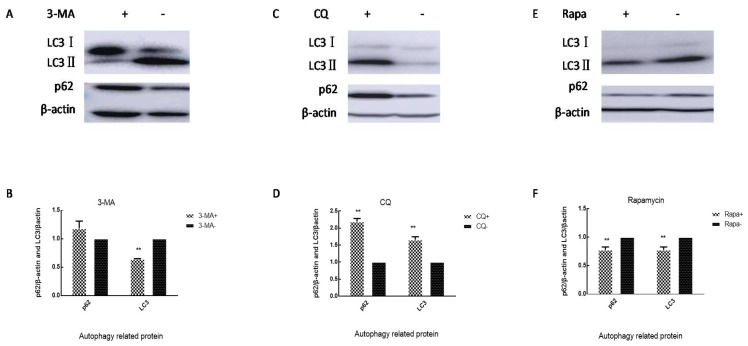
Cells were treated with 3-MA 50 μM, CQ 20 μM, and rapamycin 10 nM. After 2 h, cells were washed three times, and the levels of p62 and LC3 in the cells were measured after 24 h. (**A**) Western blot detects the effect of 3-MA on autophagy in SH-SY5Y cells. (**B**) Changes in intracellular p62 and LC3 levels in SH-SY5Y cells after 3-MA treatment of cells. (**C**) Western blot detects the effect of CQ on autophagy in SH-SY5Y cells. (**D**) Changes in intracellular p62 and LC3 levels in SH-SY5Y cells after CQ treatment of cells. (**E**) Western blot detects the effect of rapamycin on autophagy in SH-SY5Y cells. (**F**) Changes in intracellular p62 and LC3 levels in SH-SY5Y cells after rapamycin treatment of cells. The data represent means + SD from three experiments, ** *p* < 0.001, compared with the control group.

**Figure 6 viruses-12-00011-f006:**
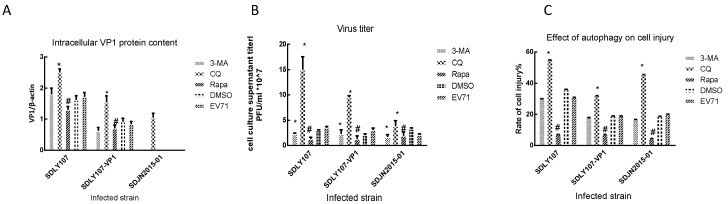
(**A**) Intracellular VP1 protein content. SH-SY5Y cells were treated with 3-MA, CQ, or rapamycin for 2 h and then infected with SDLY107, SDLY107-VP1, and SDJN2015-01, respectively. Cell lysates were harvested after 24 h. LC3, p62, and capsid protein VP1 levels were detected by western blot. (**B**) Virus titer of culture supernatant. SH-SY5Y cells were treated with 3-MA, CQ, or rapamycin for 2 h and then infected with SDLY107, SDLY107-VP1, and SDJN2015-01, respectively. After 24 h, the cell culture supernatant was harvested and the culture supernatant titer was measured by plaque assay. (**C**) Effect of autophagy on cell injury. SH-SY5Y cells were treated with 3-MA, CQ, or rapamycin for 2 h and then infected with SDLY107, SDLY107-VP1, and SDJN2015-01, respectively. After 24 h, LDH activity was assayed using the Cytotoxicity Assay Kit according to the manufacturer’s protocol. Rapa stands for rapamycin. The data represent means +SD from three experiments. * *p* < 0.05, compared with the EV71-treated group; # *p* < 0.05, compared with the DMSO-treated group.

**Figure 7 viruses-12-00011-f007:**
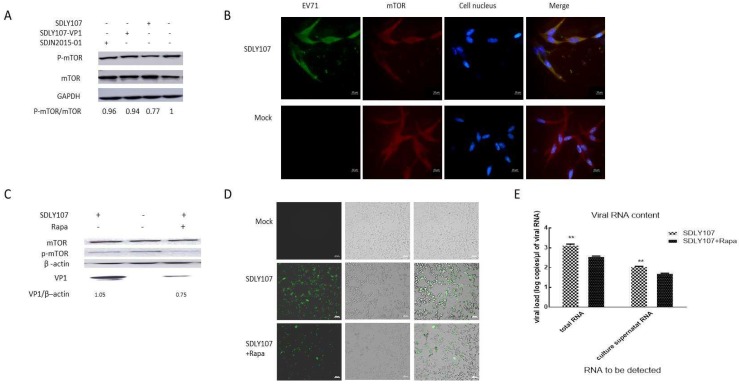
**(A)** SH-SY5Y cells were infected with the three strains. Cell lysates were collected at 24 h. mTOR and p-mTOR levels were detected by western blot. (**B**) EV71 and mTOR were detected by co-localization immunofluorescence at 24 h after infection. FICT stained EV71 green, TRITC stained mTOR red, and DAPI stained nucleus blue. (**C**) SH-SY5Y cells were seeded on a six well plate and exposed to rapamycin or not before being infected with EV71. The viral VP1 protein was examined by western blot after 24 h. (**D**) SH-SY5Y cells were inoculated into a 12 well plate with about 2 × 10^5^ cells per well and exposed to rapamycin or not before being infected with EV71. Immunofluorescent assay was used to detect the intracellular virus content after 24 h. (**E**) SH-SY5Y cells were inoculated onto a 12 well plate and exposed to rapamycin or not before being infected with EV71, and the amount of viral RNA was measured by real-time fluorescent quantitative PCR after 24 h. Rapa stands for rapamycin. The data represent means + SD from three experiments, ** *p* < 0.001, compared with the rapamycin treatment group.

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
