# Peer review of "Enterovirus 71 VP1 Protein Regulates Viral Replication in SH-SY5Y Cells via the mTOR Autophagy Signaling Pathway"

_viruses, 2019, doi:10.3390/v12010011_

Round 1
Reviewer 1 Report
My comments: In the manuscript by Liu et al., the authors investigate the relationship between the structural protein VP1 and autophagy. The major comment concerns lack of precisions and analyses :
Point 1. - Line L18-L19: “A recombinant virus SDLY107-VP1 was rescued based on VP1 coding region of EV71 severe strain was replaced and tested for replication and virulence.” - L79-L80: “In this study, a recombinant virus that replaced the VP1 fragment was constructed to explore the effect of VP1 on virus replication.” Precise the region and the viral strains used for this change. Precise also the differences between VP1 sequence for these EV71 strains.
Point 2. - Line L101-L102: “In vivo and in vitro experiments show that the strain has faster replication and stronger pathogenicity [25,26].” Precise which strains are compared in this sentence. Ref 25 and ref 26 do not correspond to the strains used by Liu et al.
Point 3. Fig. 3 A - Precise whether replication in RD cells is similar or different to viral replication in neural cells. - Precise also which viral RNA is quantified by qRT-PCR (plus or minus strand, both). - Precise the number of experiments done, it is mentioned “in triplicate”, mention how many times. - Statistical data have to be added.
Point 4. Fig 3B-3C : - Precise the time points post infection. - The authors have to define “cell survival”, “cell injury”, “cell proliferation”, “cell viability” and precise which tests they used to evaluate them then use the same term all along the ms (M & M, results, figure and figure legends). If cell injury (Fig. 3B) is the inverse of cell survival (Fig. 3C), remove one of them.
Point 5. - Fig 4A : Precise if SDLY107 strain infection results in a significant increase or not in LC3 content. Idem for SDJN2015-01 and SDLY107-VP1 strains.
Point 6. Line 218-219 : “The induction of autophagy of SDLY107-VP1 strain is less pronounced than that of reference SDLY107 strain”. - The authors must quantify infected cells and autophagy, then perform statistical tests before concluding. - The quality of image of Fig. 4B have to be increased.
Reviewer 2 Report
The authors have addressed all the comments and suggestions I made in my previous review. The quality of the article has improved. However, there are still some important points that need to be clarified or fixed.
We here summarize these points:
Page 3 line 96-97+ page 4 line 163-164. These two sentences seem to describe the same. There is no reference for the in-house developed mouse polyclonal antibody used in the study. Otherwise, the authors should include more information regarding antibody validation.
Page 3 line 107. … was isolated from a fatal patient …… …. This should be …was isolated from a fatal case of HFMD …..
Page 3 line 136. SH-SY5Y cells were inoculated into 12-well plates at a cell density of 70-80% per well and infected with EV71 strains at MOI =1. This sentence is unclear. This should be… SH-SY5Y cells were cultured (or seeded) at a density of (specify density in cells per cm2 or cells per ml) in a 12-well plates and, when SH-SY5Y reach 70-80% percent confluence, cells were infected with EV71 strains at MOI =1.
Page 4 line 145. SH-SY5Y cells were seeded in a 96-well plate at 70%–80% confluency and infected with each strain of EV71. As mentioned above, this sentence is unclear.
Page 6 Lines 224-228. While no significant difference among the replication curves of the three strains of virus could be observed in RD cells (Figure 3A), the replication power of the recombinant virus SDLY107-VP1 was significantly weaker than that of the parent strain SDLY107 in SH-SY5Y cells (Figure 3B). Accordingly, the authors conclude that replacement of VP1 gene reduces viral replication in neurons. In terms of mechanisms and processes that might be important to understand the role that VP1 plays in the replication cycle of EV71 in neuron cells, I encourage the authors to provide a possible explication as to why replacement of VP1 gene reduces viral replication in neurons, but not in non-neuronal cells like RD.
Page 6. We observe that in the manuscript, three figures (Figure 4C, Figure 4D and Figure 4E) are not labelled. Kindly provide appropriate labels for those figures in Results, in the subsection 3.3 headed “ Autophagic flux in SH-SY5Y cell is inhibited by EV71 VP1 protein.
Page 7. There are some inconsistencies between Figure 4A and Figure 4C. The result derived from the quantitative analysis of protein levels of p62 in cell lysates collected at 24h after infection with SDLY107 (Figure 4C) does not match what appears on Fig 4A.
Page 8 line 278-279. The culture supernatant was collected after 24h and subjected to plaque assay to determine the viral titer. The methods do not state how virus titres was measured using a plaque assay. This omission makes it harder for reviewers to establish the likely reliability of the results and for researchers to reproduce the experiments.
Page 9. Figure 6 is not clear enough to give meaningful information on level of LC3 and p62 detected by western blott. In addition, the authors neglected to include results regarding VP1 levels in SH-SY5Y cells infected with SDJN2015-01. Therefore, Figure 6 is difficult to interpret and adds little. Consider deleting it (making the key points within the text). A new figure showing the levels of VP1 in SH-SY5Y cells that were treated with 3-MA, CQ, rapamycin for 2h and then infected with SDLY107, SDLY107-VP1, and SDJN2015-01, should be included in the multipanel figure 7. Please note that the former Figure 7 will be now labelled as Figure 6. Likewise. Figure 8 in the present manuscript will be labelled as Figure 7.
Round 2
Reviewer 1 Report
The authors have significant improved their manuscript
This manuscript is a resubmission of an earlier submission. The following is a list of the peer review reports and author responses from that submission.
Round 1
Reviewer 1 Report
In this article, Liu et al have generated an infectious and recombinant E71 cDNA clone by replacing the VP1 gene fragments of the EV71 severe strain SDLY107 with the VP1 gene of the EV71 mild strain SDJN2015-01. By using this approach, the authors explore the effect of VP1 on EV71 replication in human neuroblastoma SH-SY5Y cell line. They propose a novel role for the EV71 VP1 protein in the disruption of autophagy, which could influence the replication of EV71 in human neurons. The molecular mechanism described in the manuscript is interesting, but that at this point the conclusions are not sufficiently supported by the data.
Major comments (line numbers are specified by L followed by the number of the line).
1- The manuscript is difficult to read. The manuscript would benefit from a through edit for grammar and word usage.
2- L108-109. Please provide more details about how the EV71 stocks were prepared and titrated.
3- L111. Please specify which cells were used for infection with SDLY107, SDJN2015-01, and SDLY107-VP1. I assume SH-SY5Y cell line.
4- L112. Did the authors assess other MOIs before choosing 1?
5- To measure viral replication kinetics, SH-SY5Y was infected with EV71 strains and the virus growth was measured by determining viral RNA (qRT-PCR) in the supernatant of viral-infected cells at different time points post-infection. It remained a bit unclear to me why did the authors choose this approach. The authors may be better served in considering other methods that measure amount of infectious virus particle by plaque assay (PFU/mL) or by end point dilution (TCID50).
6- Although the extracellular virus release determined by qRT-PCR was lower in SH-SY5Y cells infected with SDJN2015-01 and SDLY107-VP1 strains than that of the parental strain SDLY107, the intracellular virus production cannot be completely ruled out. Thus, intracellular viral production after controlled thawing, vortexing, and centrifugation must also be studied in cell pellets in order to have a reliable measure of viral replication kinetics and replication ability of the recombinant virus.
7- The viral load at 0h (~ 2-3 log copies/uL of viral RNA) shown in Figure 3A is very difficult to interpret. Does this mean that supernatant was harvested before the virus had been washed off? What is the detection limit of the assay?
8- It is not clear whether cell viability was monitored over the course of infection in parallel with the replication kinetics study. Thus, it is not possible to draw any reliable conclusion on the viral titer and cell injury rate at different stage of infection.
9- The authors performed assays measuring leakage of the intracellular enzyme LDH into extracellular space. They demonstrated an increase in viral titer in the supernatant, especially between 84 hours-120 hours, with no cell destruction. This evidence should be confirmed by other viability tests, which are complementary to LDH.
10- How can these viruses egress SH-SY5Y cells without breaching the plasma membrane integrity? One possible explanation is that these viruses (i.e., SDJN2015-01, and SDLY107-VP1) are released from infected neuronal cells within vesicles which may have bypassed lysosomal degradation, an exit mechanism termed Autophagosome-mediated exit With Out Lysis (AWOL). However, neither SDJN2015-01 nor SDLY107-VP1 induced Autophagy in SY5Y cells. Surely the authors should provide an explanation for this.
11- L220-221/L307-308. The authors argue that SDLY107 could induce autophagy in SH-SY5Y cells and inhibit the degradation of autophagosomes based on the evidence that SDLY107 infection results in an increase in LC3 content. This is over-stating the association. Autophagy is a dynamic process where autophagosomes are continually formed and degraded; the accumulation of autophagosomes could result from increased formation, decreased maturation and autophagosome turnover, or reduced fusion with lysosomes. The authors should reword this to allow for the different possibilities. In addition, Western blot analysis revealed that protein levels of p62 (markers for autophagy-mediated protein degradation), remained unchanged in SH-SY5Y cells infected with SDLY107 compared to cells infected with SDJN2015-01, SDLY107-VP1 or mock-infected cells. I don’t see how the virus inhibits the degradation of autophagosomes without affecting p62 protein levels.
12- L239-241. The titer of three strains of the virus in the rapamycin treated group was significantly reduced compared with that of the control group. This result is in contrast with previous studies which have found increasing EV71 virus titers in the culture supernatant of motor neuron cell lines and human neuroblastoma in the presence of increasing doses of rapamacyn (A-Too IH, Yeo H, Sessions OM, et al. (2016) Enterovirus 71 infection of motor neuron-like NSC-34 cells undergoes a non-lytic exit pathway. Sci Rep 6: 36983. 10.1038/srep36983. B- Huang SC, Chang CL, Wang PS, Tsai Y, Liu HS. Enterovirus 71-induced autophagy detected in vitro and in vivo promotes viral replication. J Med Virol. 2009 Jul;81(7):1241-52. doi: 10.1002/jmv.21502). Rapamycin targets the major negative regulator of autophagy; the mammalian target of rapamycin (mTOR), thus inducing the formation of autophagosomes. Thus, the decrease in viral titer when autophagy was promoted by rapamacyn (L313-314) do not support that induction of autophagy during EV71 infection of human neurons enhance the production of infectious viral particle and that autophagosomes could not serve as a scaffold on which the viral replication machinery is assembled, as described in many articles. This makes it difficult to evaluate the biological relevance of the present findings.
13- L-253. The authors state that VP1 regulates viral replication and cell injury by affecting autophagy, but they do not provide experimental support to their conclusions. This is a major drawback of the study.
Author Response
Dear Reviewer:
Thank you for your comments concerning our manuscript entitled “Enterovirus 71 VP1 protein regulates viral replication in SH-SY5Y cells via the mTOR autophagy signaling pathway” (ID: viruses-571342). All your suggestions are very important. Those comments have important guiding significance for our thesis writing and scientific research work. We have studied comments carefully and have made correction which we hope meet with approval. Revised portion are marked in red in the paper. The responds to the reviewer’s comments are as follow:
Point 1: The manuscript is difficult to read. The manuscript would benefit from a through edit for grammar and word usage.
Response 1: The manuscript has been modified by a native English speaker
Point 2: L108-109. Please provide more details about how the EV71 stocks were prepared and titrated.
Response 2:
We have provided details about how the EV71 stocks were prepared and titrated (L11-L127).
Point 3: L111. Please specify which cells were used for infection with SDLY107, SDJN2015-01, and SDLY107-VP1. I assume SH-SY5Y cell line.
Response 3:
The SH-SY5Y cell line was used for infection with SDLY107, SDJN2015-01, and SDLY107-VP1 (L129).
Point 4: L112. Did the authors assess other MOIs before choosing 1?
Response 4:
We tried MOI of 0.1, MOI of 1, and MOI of 10. But SH-SY5Y cells are not the most sensitive cells to EV71. When MOI is 0.1, the cytopathic effect is too slow. When MOI is 10, the cytopathic effect is too fast, so we choose MOI of 1, the cytopathic effect is moderate, and the follow-up tests can be carried out normally.
Point 5: To measure viral replication kinetics, SH-SY5Y was infected with EV71 strains and the virus growth was measured by determining viral RNA (qRT-PCR) in the supernatant of viral-infected cells at different time points post-infection. It remained a bit unclear to me why did the authors choose this approach. The authors may be better served in considering other methods that measure amount of infectious virus particle by plaque assay (PFU/mL) or by end point dilution (TCID50).
Response 5:
Detection of viral nucleic acids and infectious virus particles are common methods for detecting viral content. Plaque assay (PFU/mL) and end point dilution (TCID50) have an advantage in detecting infectious virus particles. There are also a large number of studies to detect the replication kinetics of viruses by detecting viral nucleic acids (Wang SH, Wang A, Liu PP, et al. Divergent Pathogenic Properties of Circulating Coxsackievirus A6 Associated with Emerging Hand, Foot, and Mouth Disease. J Virol. 2018 May 14; 92(11). pii: e00303-18. doi: 10.1128/JVI.00303-18; Liyen Loh, Janka Petravic, C. Jane Batten, et al. Vaccination and Timing Influence SIV Immune Escape Viral Dynamics In Vivo. PLoS Pathog. 2008 Jan; 4(1):e12. doi: 10.1371/journal.ppat.0040012). In this study, we designed to determine whether VP1 is associated with viral replication and that quantitative changes in nucleic acid copy number are required. Therefore, we used RT-PCR to determine replication kinetics
Point 6: Although the extracellular virus release determined by qRT-PCR was lower in SH-SY5Y cells infected with SDJN2015-01 and SDLY107-VP1 strains than that of the parental strain SDLY107, the intracellular virus production cannot be completely ruled out. Thus, intracellular viral production after controlled thawing, vortexing, and centrifugation must also be studied in cell pellets in order to have a reliable measure of viral replication kinetics and replication ability of the recombinant virus.
Response 6:
We accept the review comments. Based on the suggestion, we adjusted the experimental protocol. In order to have a reliable measure of viral replication kinetics and replication ability of the recombinant virus, the culture supernatant and the cells are harvested together. Total viral RNA content was detected and statistically analyzed after controlled thawing, vortexing, and centrifugation (L137-138 and L223)
Point 7:The viral load at 0h (~ 2-3 log copies/uL of viral RNA) shown in Figure 3A is very difficult to interpret. Does this mean that supernatant was harvested before the virus had been washed off? What is the detection limit of the assay?
Response 7:
The corresponding viral load at 0h is the amount of RNA in the prepared virus suspension (L138-139).
The instrument used for qRT-PCR is manufactured by Roche, Switzerland, and its model is LightCycler 480 II. Its detection sensitivity enables the detection of single copy genes and linear range is 1-1010 copies.
Point 8: It is not clear whether cell viability was monitored over the course of infection in parallel with the replication kinetics study. Thus, it is not possible to draw any reliable conclusion on the viral titer and cell injury rate at different stage of infection.
Response 8:
Our experimental results showed that the difference in virus replication ability was most obvious between 36 and 84 hours. Therefore, we examined the rate of cell injury at 48h after infection (L225-227).
Point 9: The authors performed assays measuring leakage of the intracellular enzyme LDH into extracellular space. They demonstrated an increase in viral titer in the supernatant, especially between 84 hours-120 hours, with no cell destruction. This evidence should be confirmed by other viability tests, which are complementary to LDH.
Response 9:
Our experimental results showed that the difference in virus replication ability was most obvious between 36 and 84 hours. During this period, SDLY107 strain maintained a high level of replication, while SDLY107-VP1 and SDJN2015-01 had lower levels of replication (Figure3 A). We examined the rate of cell injury at 48h after infection (L225-227); therefore, SDLY107-VP1 and SDJN2015-01 have low cell injury rates. Although the replication levels of SDLY107-VP1 and SDJN2015-01 strains increased rapidly between 84 hours and 120 hours, the difference in replication ability of the three strains was not obvious. We did not detect the rate of cell injury between 84 hours and 120 hours.
Point 10: How can these viruses egress SH-SY5Y cells without breaching the plasma membrane integrity? One possible explanation is that these viruses (i.e., SDJN2015-01, and SDLY107-VP1) are released from infected neuronal cells within vesicles which may have bypassed lysosomal degradation, an exit mechanism termed Autophagosome-mediated exit With Out Lysis (AWOL). However, neither SDJN2015-01 nor SDLY107-VP1 induced Autophagy in SY5Y cells. Surely the authors should provide an explanation for this.
Response 10:
Our experimental results showed that the difference in virus replication ability was most obvious between 36 and 84 hours. Therefore, we examined the rate of cell injury at 48h after infection (L225-227). At this time, SDLY107-VP1 and SDJN2015-01 had lower levels of replication and cell injury rates (Figure3). This indicates that the release of the virus will damage the integrity of the plasma membrane. We did not detect the rate of cell injury between 84 hours and 120 hours.
Point 11: L220-221/L307-308. The authors argue that SDLY107 could induce autophagy in SH-SY5Y cells and inhibit the degradation of autophagosomes based on the evidence that SDLY107 infection results in an increase in LC3 content. This is over-stating the association. Autophagy is a dynamic process where autophagosomes are continually formed and degraded; the accumulation of autophagosomes could result from increased formation, decreased maturation and autophagosome turnover, or reduced fusion with lysosomes. The authors should reword this to allow for the different possibilities. In addition, Western blot analysis revealed that protein levels of p62 (markers for autophagy-mediated protein degradation), remained unchanged in SH-SY5Y cells infected with SDLY107 compared to cells infected with SDJN2015-01, SDLY107-VP1 or mock-infected cells. I don’t see how the virus inhibits the degradation of autophagosomes without affecting p62 protein levels.
Response 11: We accept the review comments and modify the manuscript.
We performed repeated experiments and statistical analysis of the experimental data found that infection of SDLY107 strain resulted in an increase in intracellular p62 and LC3 levels. The SDLY107-VP1 and SDJN2015-01 strains did not cause significant changes in p62 and LC3 levels after infection (L238-240, Figure 4A, 4C, and 4D).
We have changed the “inhibition of autophagosome degradation” to “inhibit the autophagic flux” (L242-244 and L357-L358).
Point 12: L239-241. The titer of three strains of the virus in the rapamycin treated group was significantly reduced compared with that of the control group. This result is in contrast with previous studies which have found increasing EV71 virus titers in the culture supernatant of motor neuron cell lines and human neuroblastoma in the presence of increasing doses of rapamacyn (A-Too IH, Yeo H, Sessions OM, et al. (2016) Enterovirus 71 infection of motor neuron-like NSC-34 cells undergoes a non-lytic exit pathway. Sci Rep 6: 36983. 10.1038/srep36983. B- Huang SC, Chang CL, Wang PS, Tsai Y, Liu HS. Enterovirus 71-induced autophagy detected in vitro and in vivo promotes viral replication. J Med Virol. 2009 Jul;81(7):1241-52. doi: 10.1002/jmv.21502). Rapamycin targets the major negative regulator of autophagy; the mammalian target of rapamycin (mTOR), thus inducing the formation of autophagosomes. Thus, the decrease in viral titer when autophagy was promoted by rapamacyn (L313-314) do not support that induction of autophagy during EV71 infection of human neurons enhance the production of infectious viral particle and that autophagosomes could not serve as a scaffold on which the viral replication machinery is assembled, as described in many articles. This makes it difficult to evaluate the biological relevance of the present findings.
Response 12:
Our results indicate that 3-MA can inhibit the formation of autophagosomes, CQ can inhibit autophagic flux and lead to the accumulation of autophagosomes, while rapamycin can promote an autophagic flux and accelerate the degradation of autophagosomes (L261-267). Although the role of autophagy in the protection from pathogen infection is probably complex, the most direct method of autophagy-dependent elimination of pathogens is engulfment of the pathogen by the autophagosome and subsequent killing of the pathogen by lysosome-autophagosome fusion. Rapamycin can promote an autophagic flux and accelerate the degradation of autophagosomes in SH-SY5Y cell, thereby accelerating the clearance of pathogens. Thus, the decrease in viral titer when autophagic flux was promoted by rapamacyn, while CQ inhibits autophagic flux, leading to the accumulation of autophagosomes hindering the process of autophagy-lysosomal degradation to clear pathogens. 3-MA inhibits the formation of autophagosomes and leads to a decrease in virus titer (Figure 5A and Figure 7A). Rapamycin accelerates the degradation of autophagosomes leading to a decrease in virus titer (Figure 5E and Figure 7A), and CQ leads to the accumulation of autophagosomes leading to an increase in virus titer (Figure 5C and Figure 7A). This suggests that the formation of autophagosomes is conducive to viral replication, and this conclusion supports previous research.
In the Enterovirus 71 infection of motor neuron-like NSC-34 cells undergoes a non-lytic exit pathway, the EV71 strains S41, C2, and MS selected by the author belong to the B4 subtype, the C2 subtype, and the B2 subtype, respectively. The EV71 strains used in our experiments belong to the C4a subtype. The NSC-34cells used by the author in this article is a hybridcell line obtained from the fusion between mouse neuroblastoma and motor neuron-enriched embryonic nouse apiinal cell. The SH-SY5Y cells used in our experiments are human nerve cells. Differences in viral genotypes and cell types may be one of the reasons for the difference.
In the Enterovirus 71-induced autophagy detected in vitro and in vivo promotes viral replication, the strains used by the author are also different from the strains we use. This may also be one of the reasons for the difference.
Point 13: L-253. The authors state that VP1 regulates viral replication and cell injury by affecting autophagy, but they do not provide experimental support to their conclusions. This is a major drawback of the study.
Response 13:
In this study, we used three strains of virus, SDLY107 strain, SDJN2015-01 strain, and SDLY107-VP1 strain. The SDLY107-VP1 strain was obtained by replacing the full-length gene of the SDLY107 strain VP1 with the VP1 full-length gene of the SDJN2015-01 strain. The SDLY107 strain and the SDLY107-VP1 strain were all identical except for the VP1 gene. Therefore, the difference in the replication power and the ability to induce autophagy between SDLY107 and SDLY107-VP1 was due to the difference in the VP1 gene. Further studies have found that the accumulation of autophagosomes is conducive to viral replication and leads to an increase in the rate of cell injury (Figure 6 and Figure 7). Therefore, we state that VP1 can regulate autophagy and thus affect viral replication and cell injury.
Special thanks to you for your comments.
Reviewer 2 Report
In the manuscript by Liu et al., the authors investigate the relationship between the structural protein VP1 and autophagy.
The major comment concerns lack of precisions and analyses :
Point 1.
Line L18-L19: “A recombinant virus SDLY107-VP1 was rescued based on VP1 coding region of EV71 severe strain was replaced and tested for replication and virulence.” L79-L80: “In this study, a recombinant virus that replaced the VP1 fragment was constructed to explore the effect of VP1 on virus replication.”
Precise the region and the viral strains used for this change. Precise also the differences between VP1 sequence for these EV71 strains.
Point 2.
Line L101-L102: “In vivo and in vitro experiments show that the strain has faster replication and stronger pathogenicity [25,26].”
Precise which strains are compared in this sentence. Ref 25 and ref 26 do not correspond to the strains used by Liu et al.
Point 3.
Fig. 3 A
Precise whether replication in RD cells is similar or different to viral replication in neural cells. Precise also which viral RNA is quantified by qRT-PCR (plus or minus strand, both). Precise the number of experiments done, it is mentioned “in triplicate”, mention how many times. Statistical data have to be added.
Point 4.
Fig 3B-3C :
Precise the time points post infection. The authors have to define “cell survival”, “cell injury”, “cell proliferation”, “cell viability” and precise which tests they used to evaluate them then use the same term all along the ms (M & M, results, figure and figure legends). If cell injury (Fig. 3B) is the inverse of cell survival (Fig. 3C), remove one of them.
Point 5.
Fig 4A : Precise if SDLY107 strain infection results in a significant increase or not in LC3 content. Idem for SDJN2015-01 and SDLY107-VP1 strains.
Point 6.
Line 218-219 : “The induction of autophagy of SDLY107-VP1 strain is less pronounced than that of reference SDLY107 strain”.
The authors must quantify infected cells and autophagy, then perform statistical tests before concluding. The quality of image of Fig. 4B have to be increased.
Point 7.
Fig.7B : Surprisingly, images show co-labelling of EV71 antigens and nucleus. This is a major problem as EV71 antigens do not target cell nucleus.
Minor points:
Precise the antibody used to detect EV71 by IF 5: add statistical analysis 6: statistical data are clearly presented: show statistical data between drugs and no drugs, and with bars between histograms. 8 B: present the results of infected cells counted with and without Rapa 8C, show statistical data for viral load with and without Rapa
Round 2
Reviewer 1 Report
The authors have followed many reviewer’s suggestions and I think the paper has improved substantially. However, I still have some concerns about the analyses. Authors claim that these results suggest that replacement of EV71 VP1 result in decrease replication of EV71 in SH-SY5Y cells by affecting cell autophagy. The conclusion stems from the fact that the recombinant virus SDLY107-VP1 strain failed to induce autophagy (L357-360…. SDLY107 could inhibit autophagic flux in SH-SY5Y cells and lead to the accumulation of autophagosomes. However, the recombinant SDLY107-VP1 strains failed to induce autophagy….). I think their conclusions should be taken with caution. The study does not provide information on the consequences of the replacement of EV VP1 in the early events involved in autophagosome biogenesis. Indeed, experiments that demonstrate the role of mTOR as a key molecule affecting EV71 replication in nerve cells were performed by using SDLY107 strain, but not the recombinant SDLY107-VP1 strains. In my opinion the results of from this study suggest that autophagosome accumulation in EV71-infected SH-SY5Y cells is mediated by an inhibitory effect of VP1 in later stages of the autophagic process, rather than affecting early events associated with autophagosome induction or formation. In fact, neither SDLY107-VP1 nor SDJN2015-01 affected LC3 and P62 level, thus suggesting that recombinant SDLY107-VP1 does not interferes with a later stage of the autophagy pathway in SH-SY5Y cells. Thus, the section 3.3 (Autophagy induced by EV71 is associated with the VP1 protein) should be revised. For example:
L235: the ability of autophagy induced by the recombinant virus
L241-242: The induction of autophagy of the SDLY107-VP1 strain was less pronounced than that of the reference SDLY107 strain
L244-245: SDLY107-VP1 and SDJN2015-01 could not induce obvious autophagy in SH-SY5Y cells.
L245-246. The replacement of VP1 has an effect on the ability of the virus to induce autophagy
I would strongly suggest the authors revise the term “induction of autophagy by SDLY107-VP1” throughout the manuscript. Otherwise, the authors must provide experimental support to their conclusion
Author Response
Response to Reviewer Comments
Dear Reviewer:
Thank you for your comments concerning our manuscript entitled “Enterovirus 71 VP1 protein regulates viral replication in SH-SY5Y cells via the mTOR autophagy signaling pathway” (ID: viruses-571342). All your suggestions are very important. Those comments have important guiding significance for our thesis writing and scientific research work. We have studied comments carefully and have made correction which we hope meet with approval. Revised portion are marked in red in the paper. The responds to the reviewer’s comments are as follow:
Point 1: Comments and Suggestions for Authors
The authors have followed many reviewer’s suggestions and I think the paper has improved substantially. However, I still have some concerns about the analyses. Authors claim that these results suggest that replacement of EV71 VP1 result in decrease replication of EV71 in SH-SY5Y cells by affecting cell autophagy. The conclusion stems from the fact that the recombinant virus SDLY107-VP1 strain failed to induce autophagy (L357-360…. SDLY107 could inhibit autophagic flux in SH-SY5Y cells and lead to the accumulation of autophagosomes. However, the recombinant SDLY107-VP1 strains failed to induce autophagy….). I think their conclusions should be taken with caution. The study does not provide information on the consequences of the replacement of EV VP1 in the early events involved in autophagosome biogenesis. Indeed, experiments that demonstrate the role of mTOR as a key molecule affecting EV71 replication in nerve cells were performed by using SDLY107 strain, but not the recombinant SDLY107-VP1 strains. In my opinion the results of from this study suggest that autophagosome accumulation in EV71-infected SH-SY5Y cells is mediated by an inhibitory effect of VP1 in later stages of the autophagic process, rather than affecting early events associated with autophagosome induction or formation. In fact, neither SDLY107-VP1 nor SDJN2015-01 affected LC3 and P62 level, thus suggesting that recombinant SDLY107-VP1 does not interferes with a later stage of the autophagy pathway in SH-SY5Y cells. Thus, the section 3.3 (Autophagy induced by EV71 is associated with the VP1 protein) should be revised. For example:
L235: the ability of autophagy induced by the recombinant virus
L241-242: The induction of autophagy of the SDLY107-VP1 strain was less pronounced than that of the reference SDLY107 strain
L244-245: SDLY107-VP1 and SDJN2015-01 could not induce obvious autophagy in SH-SY5Y cells.
L245-246. The replacement of VP1 has an effect on the ability of the virus to induce autophagy
I would strongly suggest the authors revise the term “induction of autophagy by SDLY107-VP1” throughout the manuscript. Otherwise, the authors must provide experimental support to their conclusion
Response 1: We accept the review comments. We have revised the section 3.3 as required (L234, L235, L241-242, L244-246) and revised the term “induction of autophagy by SDLY107-VP1” throughout the manuscript (L20, L26, L280, L357).
Special thanks to you for your comments. We look forward to learning more from you.
Reviewer 2 Report
L126 and 127: The same sentence is written twice
Point 3:
Response 3.1: I suggest to the authors to add the sentence: “There was no significant difference in the replication curves of the three strains of virus in RD cells (data not shown)”.
Response 3.2: As the sequence of the positive and the negative strands are not the same, it is not clear how the authors detected plus and minus strands.
Point 5:
Response 7: Cells on the left of the image have nuclei not stained by DAPI. Please, change the image such as all cells have stained nucleus.
Author Response
Response to Reviewer Comments
Dear Reviewer:
Thank you for your comments concerning our manuscript entitled “Enterovirus 71 VP1 protein regulates viral replication in SH-SY5Y cells via the mTOR autophagy signaling pathway” (ID: viruses-571342). All your suggestions are very important. Those comments have important guiding significance for our thesis writing and scientific research work. We have studied comments carefully and have made correction which we hope meet with approval. Revised portion are marked in red in the paper. The responds to the reviewer’s comments are as follow:
Point 2.Comments and Suggestions for Authors
L126 and 127: The same sentence is written twice
Response 2: We accept the review comments and deleted the duplicate statement (L124-126).
Point 3:
Response 3.1: I suggest to the authors to add the sentence: “There was no significant difference in the replication curves of the three strains of virus in RD cells (data not shown)”.
Response 3.2: As the sequence of the positive and the negative strands are not the same, it is not clear how the authors detected plus and minus strands.
Response 3: We accept the review comments and the specific responses are as follows.
We have added the sentence: “There was no significant difference in the replication curves of the three strains of virus in RD cells (data not shown)” in the section 3.2 (L224-225).
We extracted total viral RNA including positive and negative strands, however our primers for detecting viral RNA were EV71 universal primers (designed for positive strand RNA of EV71). Therefore, we tested the positive strands.
Point 5:
Response 7: Cells on the left of the image have nuclei not stained by DAPI. Please, change the image such as all cells have stained nucleus.
Response 5: We accept the review comments and changed the picture (L327-328).
Special thanks to you for your comments. We look forward to learning more from you.